# A Qualitative Systematic Literature Review of the Role of U.S. Pharmacists in Prescribing Pre-Exposure Prophylaxis (PrEP)

**DOI:** 10.3390/pharmacy11010009

**Published:** 2023-01-05

**Authors:** Alina Cernasev, Tyler C. Melton, Hilary Jasmin, Rachel E. Barenie

**Affiliations:** 1University of Tennessee Health Science Center, College of Pharmacy, 301 S. Perimeter Park Drive, Suite 220, Nashville, TN 37211, USA; 2University of Tennessee Health Science Center, College of Pharmacy, 1924 Alcoa Hwy, Box 117, Knoxville, TN 37920, USA; 3University of Tennessee Health Science Center, Health Science Library, 877 Madison Avenue, Memphis, TN 38103, USA; 4University of Tennessee Health Science Center, College of Pharmacy, 881 Madison Avenue, Memphis, TN 38103, USA

**Keywords:** PrEP, systematic literature review, pharmacist, pharmacy practice, US

## Abstract

Pre-Exposure Prophylaxis (PrEP) is an effective treatment to combat the human immunodeficiency virus (HIV) endemic, but the uptake of PrEP has been low in the United States (U.S.). While individuals may access PrEP via obtaining a prescription from their prescriber and having it dispensed by their pharmacist, less cumbersome access points may exist. This systematic literature review evaluates qualitative literature to explore the role of pharmacists, pharmacy services, and interprofessional collaborations for persons seeking PrEP in the United States. Four electronic databases (PubMed, Scopus, CINAHL, and Embase) were searched in February 2022 and yielded 3841 results. After excluding duplicates, two researchers reviewed 2461 studies. These results were screened for inclusion and exclusion criteria and yielded 71 studies for full review. Out of these 71 studies, five studies met the pre-selected inclusion criteria. Of the five studies, four were qualitative studies, and one was a mixed-methods study. The studies examined different aspects of initiating PrEP and diverse outcomes, such as screening for PrEP, barriers to access PrEP, feasibility to access PrEP, accessibility via community pharmacy to PrEP, and interdisciplinary collaboration between members of the healthcare team to expand patient access to PrEP. A gap in the qualitative literature focusing on U.S. pharmacists’ roles in initiation and provision of PrEP for diverse populations may exist. While PrEP promotion and uptake are largely affected by convenience and accessibility, future interventions and strategies should include training pertaining to PrEP screening, stigma reduction, privacy considerations, and PrEP dispensing.

## 1. Introduction

The human immunodeficiency virus (HIV) epidemic continues to have a significant negative impact on public health and the United States (U.S.). In 2019, the U.S. had an estimated 1,189,700 people living with HIV (PLWH) [1]. The Centers for Disease Control and Prevention (CDC) continually monitors new HIV diagnoses as part of the *Ending the HIV Epidemic in the U.S.* program, and, in 2019, 36,801 people received a new diagnosis [1].

The Food and Drug Administration (FDA) approval of the first Pre-Exposure Prophylaxis (PrEP) medication in 2012 (emtricitabine, tenofovir disoproxil fumarate) provided opportunities to achieve the ultimate goal of decreasing HIV transmission [2,3]. In 2019, the FDA approved another medication (emtricitabine and tenofovir alafenamide) for PrEP in men who have sex with men (MSM) and transgender women, which was shown to be non-inferior to the medication already on the U.S. market and provided an additional PrEP agent for patient use [4]. Both of these medications are taken orally, once daily and can decrease HIV acquisition risk from sexual activity by approximately 92–99% [5,6,7]. 

Since the approval of PrEP, however, the uptake of the medications has been slow. The CDC reported only 15% of people who may benefit from the medication actually took it in the United States [2]. The underutilization of PrEP in the U.S. has been attributed to several unique obstacles, such as geographical limitations, stigma, and limited healthcare capacity [8,9]. Previous studies note the usage of PrEP in marginalized populations and communities has decreased disproportionately as compared to the White U.S. population [10]. Another systematic review emphasized the multifaceted inequalities preventing adequate PrEP implementation in the U.S. [11]. These studies provide further evidence of an unmet need for HIV prevention in ethnic minorities, young adults, and women, creating an opportunity to target these groups with PrEP for HIV prevention. 

Approximately 90% of Americans live within five miles of a community pharmacy, making pharmacists a readily accessible healthcare provider [12]. Pharmacists’ accessibility, pharmaceutical expertise, and expanding roles regarding managing medication therapy under collaborative pharmacy practice agreements (CPPA) positions pharmacists to improve access to evidence-based PrEP care within U.S. communities. Pharmacists’ role in providing PrEP and improving community health is widely accepted by populations benefiting from PrEP, such as Men who have Sex with Men (MSM) [13]. Crawford et al. reported widespread acceptance by MSM for pharmacists and pharmacies to provide PrEP and reported MSM perspectives of pharmacies being more convenient and accessible compared to physician offices [13].

Prior research, however, has identified gaps in continuing professional development for practicing pharmacists pertaining to their knowledge about the use of PrEP to slow the transmission of HIV [14]. One study highlighted a global need to better equip pharmacists with the knowledge and clinical skills to combat HIV [14]. While prior recommendations focused on practicing pharmacists, educators must also recognize the need to adequately prepare student pharmacists with the knowledge and skills to act and initiate PrEP when they begin practicing pharmacy. In order to more effectively curb the spread of HIV, pharmacist education must also be provided outlining the evidence-based treatment options and legal tools available to equip pharmacists to better help their patients [14].

To date, a comprehensive, qualitative systematic review of the existing literature on U.S. pharmacists’ role in initiating and providing PrEP for diverse populations is unknown [15]. The objective of this study is to evaluate qualitative literature to further explore the role of pharmacists, pharmacy services, and interprofessional collaborations for persons seeking PrEP in the U.S.

## 2. Methods

Four electronic databases (PubMed, Scopus, CINAHL, and Embase) were searched in February 2022 using a combination of keywords, Medical Subject Headings (MeSH), and/or Emtree subject headings. A health science librarian (H.J.) designed the search strategy and conducted searches across all four databases. Primary concepts were built around terms encompassing PrEP and pharmacists’ roles. (See Appendix A for full search strategy.) All 3841 results were imported into EndNote, version 20 (Clarivate, Philadelphia, PA), and duplicate results were removed. A total of 2461 results were imported into Rayyan QCRI (Qatar Computing Research Institute, Doha, Qatar), an online platform designed to expedite the screening process, for review. The protocol for this systematic literature review was registered with the Prospective Register of Systematic Reviews (PROSPERO # 312347). PRISMA Flow Diagram (Figure 1) illustrates this process [16].

### 2.1. Study Selection, Screening, Inclusion and Exclusion Criteria

This systematic literature review focused on qualitative studies conducted in the U.S. after 2012, when the FDA first approved PrEP. The rationale for selecting only U.S. qualitative studies was the uniqueness that qualitative literature provides in capturing in-depth information about certain aspects to accessing evidence-based care.

Two reviewers (A.C. and T.M.) conducted a blind screening of 2390 results. The screening process was conducted in two phases based on the inclusion and exclusion criteria. The inclusion criteria included qualitative and mixed-methods studies describing the U.S. pharmacist’s roles in initiation and providing PrEP for diverse populations, including women, transgender, and MSM, and educational studies focused on student pharmacists. The exclusion criteria were composed of studies outside the U.S., gray literature, pharmacists’ roles in patient care for HIV treatment, policy, commentary, stewardship, and pharmacogenomics.

In the first phase of screening, reviewers examined study titles and abstracts, with 2390 results being excluded. After exclusion, the screening was unblind, and the team met to discuss any discrepancies. In the second phase, the reviewers (A.C. and T.M.) conducted a full-text review of 71 results based on the described inclusion and exclusion criteria. Conflicts between reviewers were resolved by a third reviewer (R.E.B.).

### 2.2. Data Abstraction and Quality Assessment

Two researchers (A.C. and T.M.) independently abstracted the details related to the qualitative study design, demographics of the population, intervention, co-intervention, and the main outcomes [17,18]. Both researchers independently assessed the quality of each included qualitative study using the GRADE CERQual and classified them as low, medium, or high. The GRADE CERQual Evaluation assessed the following criteria: Methodological Limitations, Relevance, and Coherence and Adequacy of data [19]. The findings of qualitative studies with similar information and outcomes measures were grouped together. Thus, this qualitative systematic review included studies using mixed methods and qualitative methods, such as interviews, focus-group discussions, or both, that were conducted in person or using online tools.

## 3. Results

A total of 3841 results were identified. Of those, 1380 were duplicate results and subsequently removed (Figure 1). A total of 2461 abstract results remained for review. After the abstracts were reviewed, 2390 did not meet the inclusion criteria. The full texts of the 71 remaining results were reviewed for inclusion. Only five studies met the inclusion criteria, including four qualitative studies and one mixed-methods study. There were no studies meeting the inclusion criteria, which addressed education, student pharmacists, and transgender patients. Each study’s qualitative design was assessed using the GRADE CERQual Evaluation approach; is the results are reported in Table 1.

One of the included studies used a theoretical framework for its study design, and none of them aimed to develop a theory [18]. The studies included examined different aspects of intervention processes and outcomes such as screening for PrEP, barriers to accessing PrEP, feasibility to accessing PrEP, accessibility via community pharmacy to PrEP, and interdisciplinary collaboration within the healthcare team to expand access to PrEP services. Collectively, four studies focused on screening for PrEP, but all studies had a major commonality of highlighting the need for PrEP education and workflow implementation training.

Two studies (Crawford et al., 2020 and Hopkins et al., 2020) examined the possibility of screening patients through pharmacies for potential initiation of PrEP [13,21]. Crawford et al., 2020, reported obstacles related to pharmacy initiation of PrEP, such as privacy and adequate staff training, and highlighted the importance of identifying legislation at the state level hindering or preventing pharmacies from providing screening services for PrEP [13]. The main focus of the Hopkins et al. study was to capture qualitative data from pharmacists and pharmacy technicians, and the themes uncovered revealed similar findings to Crawford et al.’s study [21]. For instance, the studies’ participants reported interests in patient screening and implementation of PrEP services via pharmacies [13]. Furthermore, the interviewed pharmacists highlighted the importance of developing a training program for the pharmacy staff to implement routine screening and eligibility requirements for PrEP services [13].

The advantages of initiating PrEP and Post- Exposure Prophylaxis (PEP) through retail pharmacies were examined by one solely qualitative study (Koester et al.) and one mixed methods study [22,23]. Koester et al. described patient accessibility as one of the main advantages for implementing PrEP and PEP in retail community pharmacies [22]. However, this advantage comes with a workflow caveat that may be difficult to incorporate in pharmacy practice settings without modification of workflow. Additionally, pharmacists must have a clear referral process to provide patients with treatment options for conditions identified during the initiation of PrEP that would otherwise be treated in a primary care setting [22]. One factor that contributed to the successful implementation of the PrEP and PEP protocols in a pharmacy setting was the development of a stigma-free work environment [23]. This was achieved through the appropriate training of pharmacy staff on how to behave when encountering HIV-positive persons and how to combat HIV through preventive measures [23].

Gregg et al. highlighted obstacles to initiating PrEP, including inadequate screening and the presence of inconsistencies in the workflow process to initiate PrEP in the Veterans Affairs (VA) system [20]. In order to implement a successful PrEP program, Gregg et al. explored the main obstacles to accessing PrEP and presented a model for integrating PrEP as part of a primary care clinic for homeless veterans [20]. The study created an institutional prescription protocol for primary care providers and clinical pharmacists to provide and initiate PrEP for patients [20].

## 4. Discussion

This review identified several key themes for the role of pharmacists in providing PrEP services. Pharmacists are accessible to patients seeking PrEP and are willing to provide PrEP services, but the implementation of additional training and education is needed within pharmacy workflow. There are several key factors affecting pharmacists’ ability to provide PrEP services including: privacy considerations, PrEP training and education, and protocol development. As evidenced by this review, few qualitative studies exist concerning the implementation of PrEP services in pharmacy practice. PrEP promotion and uptake are largely affected by convenience and accessibility, but no studies were found examining student pharmacist perceptions or beliefs on offering PrEP. It will be increasingly valuable to be able to explore student pharmacists’ perceptions through the qualitative lens to gain a complete and comprehensive portrait of the pharmacy profession. Future interventions and strategies should specifically include training pertaining to PrEP screening, stigma reduction, privacy considerations, and PrEP dispensing.

A recurrent theme among several studies included in this systematic literature review was two-fold, with enhancing privacy protections for patients and providing training to reduce stigma being identified as necessary for the successful implementation of PrEP services in pharmacy practice settings [13,22,23]. More evidence for promoting privacy in the community pharmacy setting was highlighted by Tarfa et al. in their study focusing on people living with HIV [24]. Creating a safe and affirming environment for patients to seek specialized services in pharmacy settings, such as PrEP, requires proactive approaches to ensure privacy needs are met [25]. These proactive approaches include the application of privacy strategies such as private counseling spaces, pharmacy layout modifications, and exercising professional judgement [25]. Furthermore, pharmacists also agree that additional training on how to appropriately screen and counsel patients on HIV prevention is critical for the implementation of PrEP services in pharmacy practice settings [13,20,21,22,23].

Interprofessional collaboration and the use of collaborative practice agreements are essential for the expansion of PrEP services in pharmacy practice [26]. For example, one of the studies focused on the implementation of PrEP services in a local primary care VA clinic serving homeless patients, where interprofessional and interdisciplinary cooperation was necessary for the successful implementation and sustainability of PrEP services [20]. Other studies in our search results highlighted the need for interprofessional partnerships, but specifically physician–pharmacist collaboration, which oftentimes results in a collaborative practice agreement, thereby affording pharmacists the ability to prescribe PrEP medications [13,20,21]. Additionally, Zhu et al. reported that patients who had previous encounters with pharmacists or who had previously taken PrEP were associated with having positive views towards pharmacists prescribing PrEP [27]. Zhu et al. also noted that future work should focus on patient populations that are at a greater risk for HIV, as well as examine patient satisfaction regarding PrEP currently prescribed by pharmacists [27].

To expand on this, a study conducted by Bellman et al. provided a perspective about pharmacy barriers and initiators to facilitating the implementation of PrEP furnishing within the San Francisco area [23]. This study found facilitators for implementation included pharmacist motivation, existing interdisciplinary collaborations and partnerships with clinics and health centers, and the ability to offer patient privacy in pharmacies, while barriers included lack of time and patient awareness, obtaining necessary laboratory tests, and costs to the pharmacy [23]. Even without a CPPA in place, pharmacists were willing to sell HIV and STI testing kits to compensate for the lack of laboratory services available in most pharmacies [13].

Additionally, these interprofessional collaborations can promote access to healthcare services and reinforce the importance of facilitating pharmacist referral to PrEP-prescribing physicians as needed [13,22]. Furthermore, a study provided a comparison between pharmacist and pharmacy technician perceptions on PrEP-related screening and dispensing [21]. As a whole, pharmacist and pharmacy technician opinions were similar regarding feeling comfortable in helping patients with PrEP related activities, concerns with operationalizing services into workflow, willingness to perform interprofessional collaboration activities, and identifying time and resources as barriers to implementing PrEP services in pharmacies [21]. However, pharmacy technicians reported more concerns with privacy and implementing PrEP screening into pharmacy workflow and indicated a willingness to provide PrEP services if additional training was provided [17]. Additionally, pharmacists reported support staff needing additional training to support the screening and dispensing of PrEP [21].

Convenience and accessibility are two important factors that support the ability of pharmacists to successfully implement PrEP services in practice [13,21]. Community pharmacies are ideally positioned to provide access to PrEP services, considering 90% of patients in the U.S. live within 5 miles of a community pharmacy [28]. Cohen’s structural access theory suggests that increasing access to PrEP services is paramount for promoting community uptake of PrEP by patients [29]. Furthermore, pharmacy technicians reported the need for the community to support and raise awareness for pharmacies offering PrEP screening [13]. The access to care that pharmacies can provide, in conjunction with reducing stigma about HIV implementation of PrEP screening and training, may promote uptake and demand for PrEP dispensing in the pharmacy practice setting.

## 5. Limitations

Since the research question resulted in a limited number of qualitative studies conducted in the U.S., this review took the form of an exploratory systematic mapping and appraisal of U.S. qualitative studies. Also, due to the heterogeneity of the five studies included, it was difficult to synthesize and determining commonalties between these studies, which resulted in the inability to extract themes. This review included only English-language studies, so literature in other languages was missed. Finally, this review excluded gray literature.

## 6. Conclusions

The major finding from this review is the need for pharmacist PrEP education and pharmacy workflow-implementation training. Future qualitative studies exploring the implementation of PrEP services in pharmacy practice should include the perspectives of pharmacy technicians and pharmacists, as each is necessary for implementing and operationalizing PrEP screening and dispensing. While PrEP promotion and uptake are largely affected by convenience and accessibility, future interventions and strategies should include training pertaining to PrEP screening, stigma reduction and privacy considerations, and PrEP dispensing. As a gap in the qualitative literature suggests, future work focusing on pharmacy school curricula and pharmacy students’ perceptions is needed to more holistically understand the hesitancies and facilitators regarding the offering of pharmacy-based PrEP services.

## Figures and Tables

**Figure 1 pharmacy-11-00009-f001:**
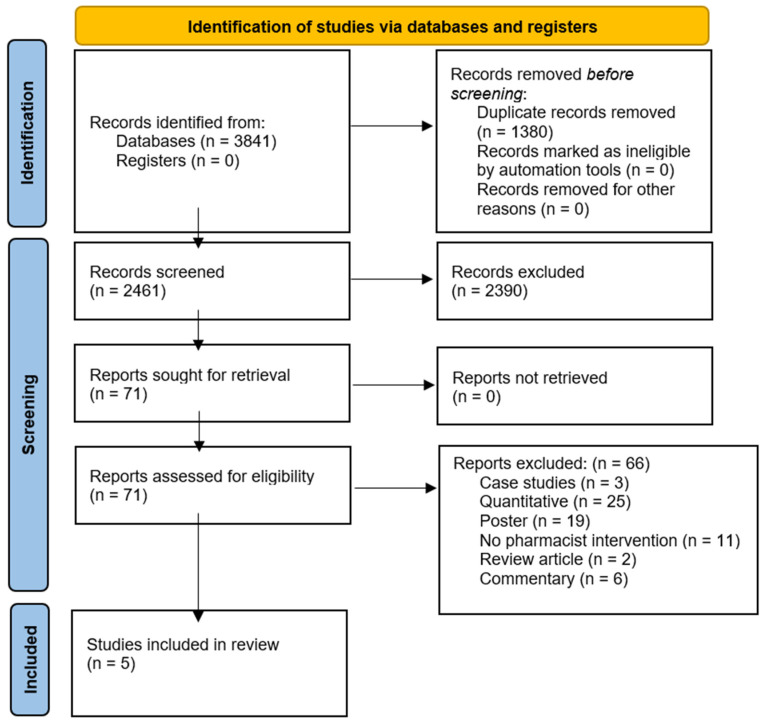
PRISMA Flow Diagram.

**Table 1 pharmacy-11-00009-t001:** GRADE CERQual Evaluation Assessment.

Author and Year	Qualitative Study Design: Interview or Focus Group	Framework	Population(s) Sampled	Location (State)	CERQual Grade
Crawford et al., 2020 [13]	Interview	N/A	Black MSM and Pharmacists	Georgia	Low
Gregg et al., 2020 [20]	Focus group	N/A	Multidisciplinary teams	California	Low
Hopkins et al., 2020 [21]	Interviews	CFIR	Pharmacists and Pharmacy Technicians	Georgia	High
Koester et al., 2020 [22]	Interviews	N/A	Pharmacists	California	Medium
Bellman et al., 2022 [23]	Interviews (Mixed methods used)	N/A	Pharmacists	California	Medium

N/A = Not available; MSM = Men who have Sex with Men; CFIR = Consolidated framework for implementation research.

## Data Availability

The data presented in this study are available upon request from the corresponding author at tmelto11@uthsc.edu.

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
