# Peer review of "A Qualitative Systematic Literature Review of the Role of U.S. Pharmacists in Prescribing Pre-Exposure Prophylaxis (PrEP)"

_pharmacy, 2023, doi:10.3390/pharmacy11010009_

Round 1
Reviewer 1 Report
This is a valuable subject and one that clearly needs to be evidence-based, in terms of practice. Therefore, a systematic review is justified, and qualitative research is valuable.
The paper (or study) does not appear to be grounded on a clear methodology for systematic reviews of qualitative data. Whilst there is some reference to standards/frameworks, this is a serious concern. This is then evident in the way that the findings are presented. The findings are not presented clearly, and this section is far too brief.
The paper is confusing to read, and I kept getting confused as to the focus, particularly around 'student pharmacists', and then later the curriculum was mentioned. So from the title, throughout the remainder of the paper, there needs to be greater clarity as to the objectives of the paper and of the study within.
There are many grammatical errors throughout and the paper must be thoroughly proofread. Similarly, parts of the methods include actual results, and this needs to be changed.
The limitations section 5 needs amending. This confuses the findings from the review, with the methods for the review. The limitations section is usually focused just on the methods.
The Conclusion is not based on the findings of the review - and includes unsupported statements.
Figure 2 was missing/not sent.
Other areas for attention:
The title is not a good representation of what the actual question is.
Abstract - needs further proofreading for grammar. Also, see line 28 - should be 'specifically' not 'specific'
Line 29 - does this mean that pharmacy students ought to be able to dispense PrEP? it's not very clear.
Line 35 - it is Centre, not Centers
Line 43 - what is the value of taking emtricitabine and tenofovir compared with previous treatment - if it is inferior, what benefit does it then have?
Line 44 - should be 'sexual activity not just 'sex'
Line 47 - should be 'reported', not 'reporting'
Line 48 - this is repetitive and covered earlier in the paper
Line 52 - what minorities are you referring to as you have then specifically stated ethnic populations - so who else?
Line 59 - need to evidence these potential benefits - e.g. how many pharmacy outlets are there, and what evidence is there that they are easily accessed in practice?
Line 75 type is 'literature'
Line 107 - why was the final stage unblinded? This is confusing because then states that it was blinded again. - Lines 106-112 are confusing. This also includes numbers that are actually part of the results, so need to be moved.
Line 115 - why and how were they grouped together and what methods were used? This is a key weakness of the paper/study in that it does not seem to be grounded on any evidence-based methods for reviewing qualitative studies.
Figure 2 was missing
Author Response
This is a valuable subject and one that clearly needs to be evidence-based, in terms of practice. Therefore, a systematic review is justified, and qualitative research is valuable.
The paper (or study) does not appear to be grounded on a clear methodology for systematic reviews of qualitative data. Whilst there is some reference to standards/frameworks, this is a serious concern. This is then evident in the way that the findings are presented. The findings are not presented clearly, and this section is far too brief.
Response: We are grateful to this suggestion. Our study followed the PRISMA guidelines where a librarian was instrumental in conducting the search strategy.
The paper is confusing to read, and I kept getting confused as to the focus, particularly around 'student pharmacists', and then later the curriculum was mentioned. So from the title, throughout the remainder of the paper, there needs to be greater clarity as to the objectives of the paper and of the study within.
Response: Thank you for this clarification. Although we searched for qualitative studies focused on student pharmacists, there were no studies conducted. Thus, it is vital to emphasize this area for future studies.
There are many grammatical errors throughout and the paper must be thoroughly proofread. Similarly, parts of the methods include actual results, and this needs to be changed.
Response: Thank you for this clarification. We addressed the grammatical errors.
The limitations section 5 needs amending. This confuses the findings from the review, with the methods for the review. The limitations section is usually focused just on the methods.
Response: Thank you for this suggestion. We have modified the limitations section to solely focus on methodology.
The Conclusion is not based on the findings of the review - and includes unsupported statements.
Response: Thank you for this suggestion. We provided more of a summary of the main findings from the study and made appropriate changes in the conclusion section.
Figure 2 was missing/not sent.
Response: Thank you for bringing this to our attention. This was an oversight, and the Figure 2 placeholder was removed.
Other areas for attention:
The title is not a good representation of what the actual question is.
Response: Thank you for this suggestion. We have amended the title to more closely match our research question.
Abstract – needs further proofreading for grammar. Also, see line 28 – should be ‘specifically’ not ‘specific’
Response: Thank you for this comment. Grammatical changes were made to the entire manuscript, and ‘specific’ was changed to ‘specifically’ in line 28.
Line 29 - does this mean that pharmacy students ought to be able to dispense PrEP? it's not very clear.
Response: Thank you for this comment. We apologize for the confusion and the sentence was revised to remove pharmacy student emphasis.
Line 35 - it is Centre, not Centers
Response: Thank you for this suggestion. It seems there is some confusion regarding the spelling. We are using the American English.
Line 43 - what is the value of taking emtricitabine and tenofovir compared with previous treatment - if it is inferior, what benefit does it then have?
Response: Thank you for this suggestion. We included further explanation about the emtricitabine and tenofovir alafenamide being an additional PrEP agent that is not inferior to emtricitabine and tenofovir disoproxil fumarate.
Line 44 - should be 'sexual activity not just 'sex'
Response: Thank you for this comment. ‘Sex’ was changed to ‘sexual activity’.
Line 47 - should be 'reported', not 'reporting'
Response: Thank you for this comment. The changes were made from ‘reporting’ to ‘reported’.
Line 48 - this is repetitive and covered earlier in the paper
Response: Thank you for this comment. This statement was removed.
Line 52 - what minorities are you referring to as you have then specifically stated ethnic populations - so who else?
Response: Thank you for this comment. This sentence was modified and includes marginalized populations and communities.
Line 59 - need to evidence these potential benefits - e.g. how many pharmacy outlets are there, and what evidence is there that they are easily accessed in practice?
Response: Thank you for this comment. This sentence was revised and includes additional information on Americans’ accessibility to community pharmacies, as 90% live within 5 miles of a pharmacy. The appropriate citation was provided as well.
Line 107 - why was the final stage unblinded? This is confusing because then states that it was blinded again. - Lines 106-112 are confusing. This also includes numbers that are actually part of the results, so need to be moved.
Response: Thank you for this concern. It was an error that there were three stages of review. There was two: title/abstract review, then full-text review. Between these review stages, the review votes are unblinded so they can arbitrate differences.
Line 115 - why and how were they grouped together and what methods were used? This is a key weakness of the paper/study in that it does not seem to be grounded on any evidence-based methods for reviewing qualitative studies.
Response: Thank you for this concern. After each study was read and measured against preset inclusion criteria, they were thematically grouped to interpret commonalities. The COREQ tool was used to qualitatively assess the data, as noted in line 126.
Figure 2 was missing
Response: Thank you for bringing this to our attention. This was an oversight, and the Figure 2 placeholder was removed.
Reviewer 2 Report
If the conflict happens between AC and TM, how the resolve the conflict will be? would you explain please?
How were you assessed the validity and the quality of the selected articles? which tool you used?
You need to re-write the citations according to MDPI submission guide please
Line (169-170: stated that: no studies were found examining student pharmacist perceptions or beliefs on offering PrEP). Do you think the pharmacy student have enough experience and knowledge to express their beliefs and opinions about PrEP? If so, is that value enough to consider it?
The authors did not provide any comparison tables compared the selected studies.
The results completely pointless and I did not see where the main result
The discussion not supportive and talk about other evidence
The goal and the main target from this research are completely shadow
Author Response
If the conflict happens between AC and TM, how the resolve the conflict will be? would you explain please?
Response: Thank you for this clarification. The conflict between AC and TM was resolved by REB. We updated the manuscript.
How were you assessed the validity and the quality of the selected articles? which tool you used?
Response: We assessed the risk of bias by using COREQ criteria.
You need to re-write the citations according to MDPI submission guide please
Response: Thank you for bringing this to our attention. The references and citations were revised to match MDPI requirements.
Line (169-170: stated that: no studies were found examining student pharmacist perceptions or beliefs on offering PrEP). Do you think the pharmacy student have enough experience and knowledge to express their beliefs and opinions about PrEP? If so, is that value enough to consider it?
Response: Thank you for this comment. Our intention here is to report that none of the studies addressed or focused on pharmacy student perceptions of PrEP. The discussion section was revised and clarified concerning student pharmacists.
The authors did not provide any comparison tables compared the selected studies.
Response: Thank you for this comment. There is not a comparison table as none of the studies were similar enough to provide comparisons.
The results completely pointless and I did not see where the main result
Response: Thank you for the comment. The manuscript was revised, and the main results of the study are clarified better.
The discussion not supportive and talk about other evidence
Response: Thank you for the comment. The manuscript underwent revision, and the discussion section was clarified.
The goal and the main target from this research are completely shadow
Response: Thank you for the comment. The main objective of this study is revied in the introduction. A summary of the study has been included in the conclusion section as well.
Reviewer 3 Report
The aim of this article is to explore the role of pharmacists, pharmacy services, and interprofessional collaborations for persons seeking PrEP in the US study by using Four electronic data- bases (PubMed, Scopus, CINAHL, and Embase).
The methodology used is well documented.
The reviewer wonders about the following facts:
From the 3841 primary results then from the 2461result after elimination of duplicates, only 5 studies were retained after application of the inclusion and exclusion criteria. We can wonder if this selection was too drastic! Drawing conclusions through these 5 studies may not be very powerful.
What do the authors think?
Despite these questions, the results obtained are interesting.

Author Response
The aim of this article is to explore the role of pharmacists, pharmacy services, and interprofessional collaborations for persons seeking PrEP in the US study by using Four electronic data- bases (PubMed, Scopus, CINAHL, and Embase).
The methodology used is well documented.
The reviewer wonders about the following facts:
From the 3841 primary results then from the 2461result after elimination of duplicates, only 5 studies were retained after application of the inclusion and exclusion criteria. We can wonder if this selection was too drastic! Drawing conclusions through these 5 studies may not be very powerful.
Response: Thank you for your time to review our manuscript. We agree with you regarding the difficulty to draw conclusions through these 5 studies.
What do the authors think?
Response: Thank you for the comment. We agree that low number of studies included in the review impact generalizability. However, this is a new and upcoming topic and little emphasis has currently been placed on qualitative research looking at the application of PrEP services in pharmacy practice settings. While the generalizability may be low, the authors feel there is still opportunity to inform implementation practices concerning PrEP services, as well as advocating for additional qualitative studies to be completed exploring this topic.
Round 2
Reviewer 1 Report
Hi, this is much improved and I am satisfied it meets the standards for publication, as set by the journal.
Regards
Roger
Author Response
Thank you for your feedback throughout these revisions. Your comments and suggestions have greatly strengthened the manuscript..
Reviewer 2 Report
Would you explain (gray literature)? based on what you judge a one literature is gray or not? Be careful this term is not acceptable in scientific papers.
Your inclusion and exclusion criteria so much superficial. Please give more details about how you select the studies. (For example: educational 104 studies focused on student pharmacists????)
You mentioned that you used (The GRADE CERQual Evaluation assessed) for each study... BUT where is the table included the grades and which questions you fixed?
You need to use JBI systematic review appraisal form for each study as well, to confirm that your inclusion and exclusion criteria are designed well.
You need TWO tables (one for GRADE CERQual and the other for JBI)
Where is the table of PICO ? You need to design it
Please give more focus on study design (in systematic way)
Author Response
Would you explain (gray literature)? based on what you judge a one literature is gray or not? Be careful this term is not acceptable in scientific papers.
- Authors Response: Thank you for this question. Gray literature is the traditional term used for research not formally published in a scientific journal in article format. Examples of gray literature include white papers and conference proceedings. It is standard practice to clarify in a methods section if gray literature was sought out or if the authors chose not to explore it.
Your inclusion and exclusion criteria so much superficial. Please give more details about how you select the studies. (For example: educational 104 studies focused on student pharmacists????)
- Authors Response: Thank you for this comment. We re-read the manuscript and are unsure of where the 104 educational studies were reported in the manuscript. However, the inclusion and exclusion criteria are referred to in the methods section, with the more information about the excluded studies being included in the PRISMA Flow Diagram.
You mentioned that you used (The GRADE CERQual Evaluation assessed) for each study... BUT where is the table included the grades and which questions you fixed?
- Authors Response: Thank you for this comment. We have included the GRADE CERQual Evaluation assessment as Table 1.
You need to use JBI systematic review appraisal form for each study as well, to confirm that your inclusion and exclusion criteria are designed well.
- Authors Response: Thank you for this comment. We have included the GRADE CERQual Evaluation assessment since this systematic review is specific to qualitative studies. We do not feel the JBI systematic review appraisal would be needed in addition to the GRADE CERQual Evaluation.
You need TWO tables (one for GRADE CERQual and the other for JBI)
- Authors Response: Thank you for this suggestion. As listed above, we have now included the GRADE CERQual Evaluation assessment as Table 1.
Where is the table of PICO ? You need to design it
- Authors Response: Thank you for this comment. Without a clinical objective to this systematic review, PICO did not seem to fit as an appropriate tool for question refinement.
Please give more focus on study design (in systematic way)
- Authors Response: Thank you for this suggestion. We further expanded on the study design by including the GRADE CERQual Evaluation Table. The study design should now be more clearly represented within the manuscript.